Effects of Chinese traditional ethnic sports on sleep quality among the elderly: a systematic review and meta-analysis

Liu Jiahui 1
Ke Xiong-Wen 1 2
Lan Yi 1
Yuan Diana 1 3
Zhang Weihao 1
Sun Jian 13659882508@163.com 1 3
1 Wushu College, Wuhan Sports University , Wuhan , Hubei , China
2 Rochester Institute of Technology , Rochester , New York , United States of America
3 Wuhan Sports University, Northeast China Ethnic Traditional Sports Research Center , Wuhan , Hubei , China
Ruiz-Montero Pedro Jesús
Electronic publication date: 2025 Feb 21
Publication date: 2025
Volume: 13
Electronic Location ID: e19019
Received 2024 Sep 27; Accepted 2025 Jan 28
Copyright: ©2025 Liu et al.
Copyright year: 2025
Copyright holder: Liu et al.
License: This is an open access article distributed under the terms of the Creative Commons Attribution License, which permits unrestricted use, distribution, reproduction and adaptation in any medium and for any purpose provided that it is properly attributed. For attribution, the original author(s), title, publication source (PeerJ) and either DOI or URL of the article must be cited.
License URL: https://creativecommons.org/licenses/by/4.0/

Keywords: Chinese traditional ethnic sports, Elderly, Sleep quality

Funding: Excellent Young and Middle-aged Science and Technology Innovation Team Program of Universities in Hubei Province: Sports Cultural Heritage and Health China T201925 This work was supported by the Excellent Young and Middle-aged Science and Technology Innovation Team Program of Universities in Hubei Province: Sports Cultural Heritage and Health China (grant numbers T201925). The funders had no role in study design, data collection and analysis, decision to publish, or preparation of the manuscript.

==============================
Objective

Sleep disorders affect approximately one-fifth of the global elderly population, with poor sleep quality among old adults linked to an increased risk of chronic conditions such as cardiovascular disease, diabetes, and hypertension. Chinese traditional ethnic sports have garnered significant attention for their potential to enhance sleep quality in older adults; however, the effectiveness of these interventions remains controversial. This meta-analysis sought to evaluate the impact of Chinese traditional sports on the quality of sleep among older adults.

Methods

A systematic search of databases, including Web of Science, Pubmed, Embase, Cochrane, China National Knowledge Infrastructure (CNKI), Wanfang, and China Science and Technology Journal Database (VIP), was conducted to identify randomized controlled trials (RCTs) examining the effects of Chinese traditional ethnic sports on sleep quality in older adults. Two independent researchers screened the literature, extracted relevant data, and conducted a quantitative meta-analysis using Stata17 software. Subgroup analyses were performed, including forest and funnel plot generation, heterogeneity analysis, sensitivity analysis, and publication bias assessment.

Results

A total of ten studies met the inclusion criteria. The meta-analysis revealed that older adults who participated in Chinese traditional ethnic sports showed significantly lower total PSQI score s relative to non-participants (Hedges’s g = −0.60, 95% CI [−0.89 to −0.31], P < 0.05). Egger’s test suggested no significant publication bias. Sensitivity analyses revealed that the age of participants, intervention mode, frequency, and duration did not substantially affect the combined effect size. Significant improvements were observed in sleep quality (Hedges’s g = −1.06, 95% CI [−2.00 to −0.13], P < 0.05), sleep duration (Hedges’s g = −0.52, 95% CI [−0.87 to −0.17], P < 0.05), and sleep efficiency (Hedges’s g = −0.50, 95% CI [−0.81 to −0.18], P < 0.05).

Conclusion

Chinese traditional ethnic sports are highly effective in improving the sleep quality of older individuals, demonstrating significant benefits in sleep quality, duration, and efficiency. Additionally, these interventions may reduce the reliance on sleep medications.

Introduction

As the global population continues to age, the prevalence of sleep disorders among the elderly is increasing, posing a significant public health challenge (Pappas & Miner, 2022; Wolkove et al., 2007). By 2050, it is estimated that individuals aged 60 and older will account for more than one-third of the world’s population (Grigsby & Olshansky, 1989). For older adults, maintaining good sleep quality is essential for preserving overall health and ensuring a high quality of life (Nelson, Davis & Corbett, 2022). Chronic sleep deprivation has been associated with a higher risk of numerous health issues, such as cardiovascular disease, diabetes, cognitive decline, and emotional disorders (Gulia & Kumar, 2018; Ma et al., 2020; Tian & Li, 2017). Addressing sleep disorders and enhancing sleep quality in the elderly has become a key focus in geriatric research. Historically, the treatment of sleep disorders has been primarily reliant on pharmacological interventions. However, conventional pharmacotherapy often presents challenges for older adults due to increased medication tolerance and heightened susceptibility to adverse side effects (Fatemeh et al., 2022; Schroeck et al., 2016). Consequently, there has been a growing interest in non-pharmacological approaches, which are now considered preferable for managing sleep disorders in this population.

With the increasing cultural exchange between China and other countries, Chinese traditional ethnic sports have garnered significant attention, especially for their potential health benefits. These sports, including Taijiquan, Wushu, and Qigong, are gaining popularity globally, with more international competitions and demonstrations emerging (Vergeer et al., 2017). Taijiquan is a traditional Chinese martial art characterized by slow, deliberate, and fluid movements that emphasize balance, relaxation, and controlled breathing. It involves a series of postures performed in a flowing sequence designed to improve flexibility, strength, and mental focus. Practitioners are encouraged to maintain a calm and mindful state, aligning their movements with deep abdominal breathing to enhance the body’s energy flow (Qi). Qigong is another traditional Chinese practice that combines movement, breathing techniques, and meditation to cultivate and balance the body’s vital energy, or “Qi”. It consists of various exercises, ranging from gentle movements to more static postures, all emphasizing slow, controlled breathing patterns. Qigong practices aim to harmonize the body and mind, alleviate stress, and promote relaxation.

Given that sleep disturbances are common among older adults and can profoundly impact health, well-being, and quality of life, understanding effective interventions is crucial. Recent studies have begun to investigate the effects of these traditional sports on sleep patterns among older individuals. Chinese traditional ethnic sports, deeply rooted in Chinese culture and characterized by their unique movement styles and philosophies, encompass a wide array of practices, such as Taijiquan and Qigong. These practices emphasize the integration of mind and body through coordinated breathing, posture, and movement, offering a holistic approach that is believed to benefit both physical and mental health.

However, the evidence regarding the impact of Chinese traditional ethnic sports on sleep in older adults remains inconsistent (Chan et al., 2014; Jahnke, Larkey & Rogers, 2010). While some studies suggest that these practices significantly improve sleep quality, reduce the prevalence of sleep disorders, and enhance both the satisfaction and sustainability of sleep, other studies failed to observe significant improvements. Given these discrepancies, this review seeks to comprehensively assess the impact of Chinese traditional ethnic sports on sleep quality in older adults (aged 55 years and above) and identify potential influencing factors. The findings aim to provide a scientific basis and theoretical support for promoting sleep health among older adults through these traditional practices.

Material and methods

The review was conducted in accordance with the PRISMA 2020 guidelines, an internationally recognized framework for systematic reviews, data inclusion, and statistical analysis (Page et al., 2021). This review was also registered with the PROSPERO International Registry of Systematic Evaluations (http://www.crd.york.ac.uk/PROSPERO/) (CRD42024548910).

Inclusion criteria

The inclusion criteria for participant selection were as follows: (1) study design: only randomized controlled trials (RCTs) were included. (2) Participants: the study population consisted of older adults aged 55 years and above. The intervention primarily targeted older adults without underlying chronic diseases, as well as those experiencing insomnia. (3) Interventions: the intervention group participated in traditional Chinese sports such as Tai Chi and fitness Qigong, while the control group engaged in regular daily activities, received health education, or took part in other low-intensity exercises. (4) Outcome measures: the primary outcome was the total score on the Pittsburgh Sleep Quality Index (PSQI), which is widely recognized as the gold standard for assessing sleep quality. The PSQI evaluates multiple dimensions of sleep, including subjective sleep quality, sleep onset latency, sleep duration, sleep efficiency, sleep disorders, etc. Its extensive validation in prior studies supports its reliability and validity, making it ideal for cross-study comparison and integration (Zitser et al., 2022). Secondary outcome measures included the following sleep-related indicators: sleep quality (McCarter et al., 2022), sleep onset (Kelly et al., 2022), sleep duration (Decat et al., 2022), sleep efficiency (Reed & Sacco, 2016), sleep disturbance (Sateia, 2014), usage of sleep medicine (Johns, 1975), and daytime dysfunction (Jackson, Howard & Barnes, 2011).

Exclusion criteria

The exclusion criteria for participant selection were as follows: (1) studies involving older adults with underlying diseases other than sleep disorders, (2) studies lacking sufficient information for data extraction, and (3) abstracts or reviews without available full-text information.

Literature search strategies

A comprehensive search was conducted across seven databases, including Web of Science (WOS), PubMed, Embase, Cochrane, China National Knowledge Infrastructure (CNKI), Wanfang, and China Science and Technology Journal Database (VIP). CNKI, Wanfang, and VIP are Chinese paper repositories. These Chinese journals are peer-reviewed and adhere to rigorous editorial processes. Additionally, many Chinese journals are now included in international databases such as Scopus and SCI. The search covered all records from the inception of each database to May 20, 2024. Both subject terms and free-text keywords were utilized in the search strategy, combining terms with Boolean operators “AND” and “OR” for precision and inclusivity. The search process underwent repeated iterations and refinement to ensure the inclusion of relevant studies.

For Chinese-language searches, the following keywords were used: “Yi JinJing”, “Wuqinxi”, “Six-Character Formula”, “Baduanjin”, “Shierduanjin”, “Twelve Methods of Fitness Qigong-Guiding Health”, “Fitness Qigong-Twelve Methods of Fitness Qigong”, “Fitness Qigong-Tai Chi Stick”, “Health Preservation”, “Tai Chi”, “Taijiquan”, “Qigong”, and “Sleep Quality”, “Quality of Life”, “Sleep Disorders”, “Sleep Phenomena”, “Insomnia”, and “Elderly”, “Middle-aged”, “Elderly”, “Senior Citizens”.

For English-language searches, the following terms were employed: “Qigong” or “Mind Body Exercise”, “Fitness Qigong” or “Traditional Chinese Exercise”, “Tai Chi” or “Taiji”, or “Baduanjin” or “Wuqinxi” or “Shierduanjin” and “Sleep Quality” or “Sleep” or “Disorders of Initiating and Maintaining Sleep” or “Insomnia” and “The Aged” or “Old Age” or “Senior Citizen ”or “Elderly ”or “Aged” or “Randomized Controlled Trial” or “Randomized” or “Controlled” or “Trial” or “RCT”. Search protocols were tailored to the unique characteristics of each database. In addition, studies included in similar meta-analyses and reviews were examined to identify and incorporate any potentially relevant literature.

Literature screening and data extraction

The process of literature screening, data extraction, and cross-checking was carried out independently by two researchers (Jia-hui Liu and Xiong-Wen Ke). In cases of disagreement, a third party (Jian Sun) was consulted to facilitate decision-making. Whenever necessary, the authors of the studies were contacted to obtain additional information or clarify missing data. The screening process began with a review of titles and abstracts to exclude any irrelevant studies. After this initial screening, full-text articles were further assessed to determine their eligibility for inclusion based on the pre-defined criteria. During the data extraction phase, the following key details were recorded, including (1) basic information about the included studies, such as the study’s title, name of the first author, nationality, publication date, and other basic details; (2) baseline characteristics of the study subjects, such as the sample size of each group and the age of the subjects; (3) specific information about the interventions, such as type of intervention (e.g., Taiji and Qigong), intervention duration categorized as ‘0–30 minutes’, ‘31–60 minutes’, or ‘61–120 minutes’, and intervention period categorized as ‘8–12 weeks’ or ‘13–16 weeks’; (4) data on dimensions of sleep, such as the total PSQI score, sleep quality, sleep onset, sleep duration, sleep efficiency, sleep disturbances, usage of sleep medications, and daytime dysfunction.

Evaluation of bias in the included studies

The risk of bias in the included studies was independently assessed by two researchers using the Risk of Bias Assessment Tool for Randomized Controlled Trials as recommended by the Cochrane Handbook. The evaluation covered seven criteria: random sequence generation, allocation concealment, blinding of subjects and investigators, blinding of assessors, incomplete outcome data, selective reporting, and other biases. The studies were classified into three levels of risk: low risk of bias (level A, studies meeting four or more of the seven criteria, suggesting a high level of methodological rigor and minimal bias risk), unclear risk of bias (level B, studies meeting two or three criteria, where there is some uncertainty regarding the methodological quality, requiring further clarification or scrutiny), and high risk of bias (level C, studies meeting one or no criteria, indicating substantial risk of bias that may affect the reliability and validity of the results). Any disagreements during the evaluation were resolved through discussion with a third researcher to ensure consensus (Cumpston et al., 2019).

Evaluation of the quality of circumstantial evidence

The quality of evidence for the outcomes was assessed using GRADEpro software. Five evaluation domains were considered: limitations, inconsistency, indirectness, imprecision, and publication bias. Each domain was independently evaluated and categorized as none (not downgraded), serious (downgraded by 1), or very serious (downgraded by 2). Based on this evaluation, the overall quality of evidence was classified into one of four levels: high, moderate, low, or very low (Li et al., 2024).

Statistical analysis

The meta-analysis was conducted using Stata17 software. Key intervention details were extracted from each study, including the timing, frequency, and duration of the exercise interventions, as well as descriptions of intervention methods and control conditions. Demographic details of the participants, such as age, gender, and insomnia status, were also recorded to facilitate subgroup analyses. To evaluate the study results, a forest plot, funnel plot, sensitivity analysis, and publication bias test were performed. Heterogeneity among studies was assessed using the I2 statistic. If I2 > 50% and P < 0.1, indicating substantial heterogeneity, a random-effects model was used, supplemented by subgroup and sensitivity analyses to explore potential sources of heterogeneity. Otherwise, a fixed effects model was used. Effect sizes were calculated using Hedges’s g, and bias correction factors were determined using precise calculations and adjusted standard errors based on Hedges’ and Olkin’s methodology. The effect sizes were categorized as small if Hedges’s g was less than 0.2, small to medium for values ranging from 0.20 to 0.49, medium for values ranging from 0.50 to 0.79, and significant for values equal to or greater than 0.8 (Yang et al., 2023).

Results

Literature search results

A comprehensive search of seven major electronic databases, including WOS, PubMed, Embase, Cochrane, CNKI, Wanfang, and VIP, yielded a total of 839 documents. After importing these into Endnote and removing 144 duplicates, 695 unique papers remained for initial screening based on their titles and abstracts. During the initial screening, 639 documents were manually excluded, leaving 56 papers for further evaluation. Of these, three articles could not be retrieved, resulting in 53 articles for re-screening using the predefined inclusion and exclusion criteria. In this phase, 44 articles were excluded for the following reasons: eight articles lacked sufficient data for extraction, 18 articles showed inconsistency in outcome indicators, 12 articles were non-RCTs, five articles were master’s theses, and one article had inconsistencies in the intervention content. Ultimately, nine articles met the inclusion criteria. An additional article was manually identified, bringing the total to ten articles included in the final review (Chen et al., 2012; Fan et al., 2020; Hosseini et al., 2011; Irwin et al., 2014; Irwin, Olmstead & Motivala, 2008; Li et al., 2004; Nguyen & Kruse, 2012; Siu et al., 2021; Wang et al., 2024; Wang, Li & Bi, 2019). The literature screening process is visually summarized in Fig. 1.

Figure 1 Literature screening process.

Basic characteristics of the included studies

The review included ten articles, of which nine were written in English and one in Chinese. The studies were conducted in diverse regions, including China, Germany, the United States, and Iran, providing a broad cultural and geographical perspective. Across the ten studies, the total sample size comprised 989 participants. Among these, seven studies focused on Taijiquan as the primary intervention, while the remaining three explored fitness Qigong. The frequency of the interventions varied from 1–3 times/week, with each session lasting 45–60 min. The duration of the intervention programs ranged from 12 to 25 weeks (Table 1).

Table 1 Basic information table.

Study	Country	N	Age	Population characteristics	Intervention mode	Intervention time	Frequency of intervention	Intervention cycle	Intervention content of the control group	
Wang et al. (2024)	China	55	68.9 ± 5.1	Normal, community-dwelling elderly people	the great ultimate	60 min	Three times/week	8 Weeks	normal life	
Siu et al. (2021)	China	181	67.3 ± 6.8	Older adults with insomnia	the great ultimate	60 min	Admidia 3 times/week	12 Weeks	normal life	
Nguyen & Kruse (2012)	Germany	96	68.9 ± 5.1	Normal, community-dwelling elderly people	the great ultimate	60 min	Two times/week	24 Weeks	normal life	
Irwin, Olmstead & Motivala (2008)	America	112	59–86	Normal, community-dwelling elderly people	the great ultimate	40 min	Three times/week	25 Weeks	health education	
Hosseini et al. (2011)	Iran	56	60–83	Normal, community-dwelling elderly people	the great ultimate	20–25 min	Three times/week	12 Weeks	normal life	
Wang, Li & Bi (2019)	China	99	71.5 ± 8.59	Normal, community-dwelling elderly people	BadJin fitness Qigong	45 min	Admidia 4 times/week	12 Weeks	Sleep knowledge preaching	
Irwin et al. (2014)	America	98	55–85	Older adults with chronic or primary insomnia	the great ultimate	120 min	1 Time/week	12 Weeks	The sleep seminar on educational control	
Li et al. (2004)	America	118	60–92	Normal, community-dwelling elderly people	the great ultimate	60 min	Three times/week	24 Weeks	Low-intensity exercise	
Fan et al. (2020)	China	119	71.1 ± 6.3	Elderly adults with community-based sleep disorders	BadJin fitness Qigong	45 min	Five times/week	24 Weeks	normal life	
Chen et al. (2012)	China	55	71.15 ± 8.13	Normal, community-dwelling elderly people	BadJin fitness Qigong	30 min	Three times/week	12 Weeks	normal life	

Quality evaluation of the included literature

The studies included in this review were all RCTs, each providing clear descriptions of their study designs and the characteristics of the participants involved. All articles reported robust data without evidence of selective reporting. However, only four studies achieved allocation concealment and researcher blinding, while assessor blinding was reported in three articles. Additionally, one article exhibited other potential biases. These findings are summarized in Fig. 2.

Figure 2 Bias evaluation.

Meta-analysis

Meta-analysis of the total PSQI score

All ten articles included in this analysis reported the total PSQI scores, revealing significant heterogeneity among studies (I2 = 78.71%, P < 0.1). The meta-analysis demonstrated a significant improvement in sleep quality associated with traditional Chinese ethnic sports among older adults (Hedges’s g = −0.60, 95% CI (−0.89, −0.31), P < 0.05). The Egger test indicated no significant publication bias (Z = 0.51, P > |z| = 0.6086). Detailed results are shown in Figs. 3 and 4.

Figure 3 Forest plot of total PSQI score.

(A) Older adults without sleep disorders; (B) older adults with sleep disorders. Studies: Wang et al. (2024), Siu et al. (2021); Nguyen & Kruse (2012); Irwin, Olmstead & Motivala (2008); Hosseini et al. (2011); Wang, Li & Bi (2019); Irwin et al. (2014); Li et al. (2004); Fan et al. (2020); Chen et al. (2012).

Figure 4 Funnel plot of the total PSQI score.

To investigate the factors influencing the effects of traditional Chinese ethnic sports on the total PSQI scores, subgroup analyses were conducted based on intervention modes, intervention time, intervention period, and participant type.

The intervention modality did not show a moderating effect on the total PSQI score (P = 0.136). However, both modalities, ‘Fitness Qigong’ and ‘Taijiquan’, demonstrated statistically significant effects, with the effect size Hedges’s g = −0.925 (P < 0.001) for the ‘Fitness Qigong’ group, and Hedges’s g = −0.476 (P = 0.006) for the ‘Taijiquan’ group.

There was no moderating effect of intervention time on the total PSQI score (P = 0.581). Duration was categorized into ‘0–45 minutes’ and ‘46–120 minutes’ groups. Both groups showed significant effects, with Hedges’ g = −0.526 (P = 0.030) for the ‘0–45 minutes’ group and Hedges’s g = −0.693 (P < 0.001) for the ‘46–120 minutes’ group. Similarly, there was no moderating effect of intervention period on the total PSQI score (P = 0.296). Periods were categorized into ‘13–36 weeks’ and ‘8–12 weeks’. A significant effect was observed only for the ‘8–12 weeks’ group, with Hedges’ g = −0.749 (P < 0.001).

The type of participant (e.g., normal older adults vs. insomniac older adults) did not moderate the total PSQI score (P = 0.388). Significant effects were found in both groups, with Hedges’s g = −0.455 (P = 0.018) for the ‘insomniac elderly’ group and Hedges’s g = −0.702 (P = 0.001) for the ‘normal elderly’ group. Table 2 shows a detailed breakdown of the results.

Table 2 Subgroup analysis table PSQI total score.

Moderator variable	N	I 2	Hedges ’g, 95% CI	P	P value
between groups	
Intervention mode					0.136	
Fitness Qigong	3	71.63%	−0.925 (−1.405, −0.445)	0.000		
Taijiquan	7	78.86%	−0.476 (−0.818, −0.133)	0.006		
Intervention time					0.581	
0–45 min	5	83.32%	−0.526 (−1.000, −0.051)	0.030		
46–120 min	5	73.41%	−0.693 (−1.050, −0.336)	0.000		
Intervention cycle					0.296	
13–36 Weeks	4	87.11%	−0.427 (−0.957, 0.103)	0.114		
8–12 Weeks	6	58.24%	−0.749 (−1.041, −0.457)	0.000		
Types of elderly					0.388	
Insomnia elderly	4	71.55%	−0.455 (−0.830, −0.079)	0.018		
Normal elderly	7	81.51%	−0.702 (−1.118, −0.285)	0.001		

Meta-analysis of sleep quality

Four studies reported data on sleep quality. However, there was significant heterogeneity among the included studies (I2 = 94.30%, P < 0.1), necessitating the use of a random-effects model for the combined analysis. The meta-analysis revealed that traditional Chinese ethnic sports had a significant and pronounced effect on improving the subjective sleep quality of older individuals compared to the control group (Hedges’s g = −1.06,95% CI [−2.00 to −0.13], P < 0.05) (Fig. 5A).

Figure 5 Forest plot showing the effects of Chinese traditional ethnic sports on the seven components of the PSQI in old adults.

(A) Sleep quality, (B) sleep onset, (C) sleep duration, (D) sleep efficiency, (E) sleep disturbances, (F) sleep medication, (G) daytime dysfunction. Studies: Siu et al. (2021); Nguyen & Kruse (2012); Irwin, Olmstead & Motivala (2008); Hosseini et al. (2011); Wang, Li & Bi (2019); Irwin et al. (2014); Li et al. (2004); Fan et al. (2020); Chen et al. (2012).

Meta-analysis of sleep onset

Six studies reported data on sleep onset. Due to the high heterogeneity in the included studies (I2 = 89.83%, P < 0.1), a random-effects model was used for the combined analyses. The results showed that Chinese traditional ethnic sports did not have a statistically significant effect on reducing sleep onset in older adults compared with the control group (Hedges’s g = −0.40, 95% CI [−0.93–0.13], P > 0.05) (Fig. 5B).

Meta-analysis of sleep duration

Six studies reported data on sleep duration. With considerable heterogeneity in the included studies (I2 = 77.19%, P < 0.1), a random-effects model was applied. The results of the meta-analysis showed that Chinese traditional ethnic sports had a significant positive effect on prolonging the sleep duration of older adults compared to the control group (Hedges’s g = −0.52, 95% CI [−0.87 to −0.17], P < 0.05). As fewer than ten studies were included, Egger’s test for publication bias was not conducted (Fig. 5C).

Meta-analysis of sleep efficiency

Six studies reported data on sleep efficiency. Due to the statistical heterogeneity in the included studies (I2 = 71.28%, P < 0.1), a random-effects model was used for the combined analyses. The results of the meta-analysis revealed that Chinese traditional ethnic sports significantly improved sleep efficiency in older adults compared to the control group (Hedges’s g = −0.50, 95% CI [−0.81 to −0.18], P < 0.05) (Fig. 5D).

Meta-analysis of sleep disturbance

Four studies reported data on sleep disturbance. Due to moderate heterogeneity among the included studies (I2 = 56.92%, P < 0.01), a random-effects model was used for the combined analysis. The meta-analysis showed that Chinese traditional ethnic sports did not have a significant effect on reducing sleep disturbance among older adults compared to the control group (Hedges’s g = −0.22, 95% CI [−0.54–0.10], P > 0.05) (Fig. 5E).

Meta-analysis of the usage of sleep medicine

Four studies reported data on the use of sleep medications. As no heterogeneity was observed among these studies (I2 = 0.00%, P > 0.1), a fixed-effects model was used for combined analyses. The results of the meta-analysis revealed a significant reduction in the use of sleep medication among older adults in the intervention group compared to the control group (Hedges’s g = −0.24, 95% CI [−0.44 to −0.03], P < 0.05) (Fig. 5F).

Meta-analysis of daytime dysfunction

Four studies reported data on daytime dysfunction. Due to high heterogeneity among the included studies (I2 = 90.34%, P < 0.1), a random-effects model was used for the combined analyses. The results of the meta-analysis showed that Chinese traditional ethnic sports did not have a significant effect on reducing daytime dysfunction in older adults compared to the control group (Hedges’s g = −0.35, 95% CI [−1.04–0.34], P > 0.05) (Fig. 5G).

Evaluation of the level of evidence

The quality of evidence for the outcomes was assessed using GRADEpro. The findings for various sleep-related outcomes, including the total PSQI score, sleep duration, sleep efficiency, sleep disturbance, use of sleep medication, and daytime dysfunction, were determined to be of moderate quality. However, evidence related to sleep onset latency and sleep duration was deemed to be of low quality due to inconsistencies and imprecision in the data. Detailed information can be found in Table 3.

Table 3 Grade of evidence for the outcome measures.

Final result
metric	RCTs/ Item	Quality evaluation of evidence	The incident happened
Times/sample size		Relative effect size	Quality	
		Boundedness	Inconsistency	Indirect	Inexactness	Publication bias	Intervention group	Control group				
Total points	10	Don’t downgrade	Don’t downgrade	Don’t downgrade	Lower the level	Don’t downgrade	505	469		−0.60 [−0.89, −0.31]	medium-quality	
sleep quality	4	Don’t downgrade	Lower the level	Don’t downgrade	Lower the level	Don’t downgrade	194	180		−1.06 [−2.00, −0.13]	Inferior quality	
Sleep time	6	Don’t downgrade	Lower the level	Don’t downgrade	Lower the level	Don’t downgrade	333	295		−0.40 [−0.93,0.13]	Inferior quality	
Hour of sleep	6	Don’t downgrade	Don’t downgrade	Don’t downgrade	Lower the level	Don’t downgrade	333	295		−0.52 [−087, −0.17]	Medium-quality	
Sleep efficiency	6	Don’t downgrade	Don’t downgrade	Don’t downgrade	Lower the level	Don’t downgrade	333	293		−0.50 [−0.81, 0.18]	Medium-quality	
Dyssomnia	4	Don’t downgrade	Don’t downgrade	Don’t downgrade	Don’t downgrade	Don’t downgrade	194	180		−0.22 [−054, 0.10]	Medium-quality	
Hypnotic drugs	4	Don’t downgrade	Don’t downgrade	Don’t downgrade	Don’t downgrade	Don’t downgrade	194	180		−0.24 [−0.44, −0.03]	Medium-quality	
Daytime function	4	Don’t downgrade	Don’t downgrade	Don’t downgrade	Don’t downgrade	Don’t downgrade	194	180		−0.22 [−054, 0.10]	Medium-quality	

Discussion

The review consolidates evidence that can guide healthcare practitioners in selecting and recommending interventions that can improve sleep outcomes among the elderly. This review indicates that Chinese traditional ethnic sports can improve the overall sleep conditions of older adults, enhancing sleep quality, increasing sleep duration, and improving sleep efficiency. These findings are consistent with previous studies (Liu et al., 2023; Saeed, Cunningham & Bloch, 2019; Yilmaz Gokmen et al., 2019). Numerous studies have highlighted a positive correlation between moderate physical activity and sleep quality. Regular exercise has been shown to regulate the biological clock, reduce sleep onset latency, and increase both the duration and depth of sleep (Bisson, Robinson & Lachman, 2019). Although aerobic exercise and strength training are also effective in improving sleep, low-intensity balance and flexibility training, such as tai chi, may be particularly beneficial for some older adults whose physical functions are gradually declining (Takemura et al., 2024). In addition to physical benefits, positive thinking meditation has been found to alleviate psychological problems such as anxiety and depression, thereby improving sleep quality by promoting mindfulness and reducing inner turmoil (Black et al., 2015). Traditional Chinese ethnic sports, as an effective intervention, can enhance sleep quality in older adults through various interrelated mechanisms, often overlapping with the effects of other physical and mental health interventions. The improvement in sleep quality through traditional Chinese ethnic sports can be explained from several perspectives. Firstly, Tai Chi increases physical activity levels, leading to higher energy expenditure and consequently, better sleep quality. Secondly, it helps alleviate negative emotions, such as anxiety, depression, and other mental health issues, which are often significant disruptors of sleep. Thirdly, Tai Chi and Qigong enhance physical stability and coordination, reducing nocturnal restlessness and promoting deeper, more restful sleep (Rogers, Larkey & Keller, 2009). Fourthly, these practices facilitate a balance between the sympathetic and parasympathetic nervous systems, promoting relaxation and an overall state of calmness (Chang et al., 2019; Yang et al., 2007). Lastly, regular participation in these exercises helps regulate the biological clock of older adults, making it easier to fall asleep and maintain a consistent sleep schedule (Jahnke et al., 2010).

The present review has identified a significant finding that traditional Chinese ethnic sports are effective in reducing dependence on hypnotic medication. This finding contrasts with the results of a previous study by Chan et al. (2016), which found no significant changes in the frequency of sleep medication use among older adults practicing Tai Chi and Qigong. However, it is important to note that Chan et al. (2016) included older adults with cognitive impairment and excluded those on medications, whereas the present review focused exclusively on older adults without underlying health conditions, including those both using and not using sleeping medications. This difference in participant selection may account for the discrepancies observed between the two studies.

Furthermore, this review found that traditional Chinese ethnic sports did not have a significant impact on sleep onset or daytime dysfunction in older adults. This finding contradicts findings from a study by Wu et al. (2021), which reviewed 22 RCTs on the effects of Chinese traditional ethnic sports and aerobic exercise on sleep quality in older adults. Wu et al.’s results suggested that Tai Chi and Qigong significantly improved sleep latency. Additionally, our findings differ from those of Li et al. (2004), who randomized participants to either Tai Chi or low-impact exercise, with one 60-minute session three times per week for 24 weeks. Li et al. reported that exercise, specifically Tai Chi, was associated with a reduction in daytime sleepiness. It is worth noting, however, that both of these previous studies specifically targeted older adults with sleep disorders, while our study included both older adults with sleep disorders and those without. Therefore, the differences in participant characteristics such as age, health status, or baseline sleep conditions, could affect the outcomes of the intervention. Additionally, variations in the duration of the intervention or intensity of participation might influence the effectiveness of traditional sports in improving sleep. These discrepancies highlight the importance of considering such factors when interpreting the results and suggest that further research, with standardized protocols, is needed to clarify the true impact of traditional ethnic sports on sleep.

Subgroup analyses of the total PSQI scores in this review highlight that both Taijiquan and Qigong are effective interventions for improving sleep quality in older adults. The gentle, slow, smooth, and continuous movements of Taijiquan contribute to enhancing muscle strength, coordination, flexibility, and balance in older adults. Additionally, the focus on breathing and mindfulness during practice (Huang et al., 2022) has been shown to significantly reduce stress and anxiety levels (Liu et al., 2020). This reduction in psychological stress plays a crucial role in alleviating sleep disturbances, such as difficulty falling asleep.

Similarly, Qigong emphasizes mental calmness and, through regulated breathing and deep relaxation, helps to balance the activities of the sympathetic and parasympathetic nervous systems. This balance induces a deeper state of physical relaxation, thereby preparing the body for sleep (Koh, 1982). The current study found that both shorter intervention times (0–45 min) and longer sessions (46–120 min) were effective in enhancing sleep quality in older adults. Taijiquan and Qigong, as forms of gentle physical and mental exercises, are not strictly dependent on session duration to yield improvements in sleep quality. Both short and longer practice sessions offer physical and mental benefits that positively influence sleep. Even brief sessions can help relax the body and alleviate physical pain and discomfort, a common factor affecting sleep quality in older adults (You et al., 2018). This flexibility and adaptability underscore the suitability of traditional Chinese exercises like Taijiquan and Qigong for older adults, making them ideal interventions for enhancing overall well-being.

Our analysis also revealed that intervention periods of 8–12 weeks effectively improved sleep quality, efficiency, and the total PSQI scores, while a longer training period of 13–16 weeks only significantly affected sleep duration. A shorter intervention period, compared to the extended periods of 13–36 weeks, may help reduce feelings of slackness and anxiety, making it easier for participants to maintain motivation. The 8–12 week duration appears sufficient for older adults to become familiar with the movements, master the techniques, and establish a consistent exercise routine. During this period, older adults are likely to remain more dedicated to their training, leading to greater effectiveness. In contrast, longer training periods may induce fatigue and stress, diminishing motivation and potentially reducing the overall impact of the intervention. Moreover, compared to older adults with insomnia, those without sleep disorders tend to have better health and adaptive capacity, making them more likely to benefit from traditional Chinese ethnic sports. In contrast, older adults with insomnia may require a more individualized and comprehensive combination of interventions to achieve optimal results.

Only four of the included studies achieved both allocation concealment and researcher blinding, while assessor blinding was present in only three studies. Furthermore, one study exhibited other potential biases, which may further influence the interpretation of its results. When allocation concealment is not properly implemented, there is a risk that the treatment groups may be imbalanced, leading to skewed or unreliable results. Additionally, the absence of researcher blinding in most studies introduces the possibility that outcomes could be influenced by the researcher’s knowledge of the treatment group, affecting their judgment and behavior. Despite the use of the Cochrane risk of bias tool, these methodological issues suggest that the findings of the studies may not fully reflect the true treatment effects. Future research should prioritize the implementation of these critical methodological safeguards to enhance the validity and generalizability of the findings. The GRADE assesment suggested low quality evidence for some key outcomes. To improve the quality of evidence in future research, several strategies can be employed. First, increasing the sample size and ensuring diverse participant inclusion can enhance the generalizability of findings. Second, ensuring rigorous methodological standards, such as consistent allocation concealment, blinding, and minimization of potential biases, will strengthen internal validity. Third, using standardized outcome measures and conducting long-term follow-up assessments can provide more reliable and comprehensive data. Lastly, conducting well-designed multicenter trials and adhering to reporting guidelines like CONSORT will help improve the transparency and reproducibility of results.

Furthermore, significant heterogeneity (e.g., I2 > 70% in some analyses) is observed. As indicated in Table 1, several factors may contribute to the significant heterogeneity observed in the analyses, including differences in the countries where the studies were conducted, participant age, and population characteristics. Additionally, variations in the intervention modes, intervention duration, frequency of intervention, intervention cycles, and the content of the control groups across the ten studies could all play a role in the observed heterogeneity. However, the results of this review underwent sensitivity analyses to assess their reliability. The findings showed that the literature used to assess the total PSQI score and sleep quality fell within the confidence intervals, indicating that the results can be considered trustworthy. However, heterogeneity was observed in sleep duration and sleep efficiency, as derived from Siu et al. (2021). Upon removing these articles from Siu et al. (2021) and re-analyzing the data, the heterogeneity in sleep duration decreased significantly (I2 = 67.38%, P < 0.1). It is important to note that Siu et al. (2021) primarily focused on older adults with moderate sleep disturbances who had not yet reached the threshold for diagnosable insomnia. This inclusion criterion may have introduced bias, as it combined older adults without insomnia disorders with those who had insomnia in both the intervention and control groups. Furthermore, the primary outcome variable in this study was self-assessed sleep quality, which should be considered when interpreting the results.

Study strengths and limitations

Strengths

This meta-analysis provides a comprehensive evaluation of the impact of Chinese traditional ethnic sports on the sleep quality of older adults, considering various intervention methods, durations, and cycles. This multi-faceted approach enhances the reliability and robustness of the findings. Given the global trend of an aging population, the focus on older adults in this study is particularly significant. It addresses the widespread issue of sleep disorders and offers the potential for reducing the prevalence of various chronic diseases in the elderly worldwide. Furthermore, Tai Chi and Qigong present valuable non-pharmacological alternatives for individuals seeking alternative or complementary treatments for sleep disturbances. The ease with which these exercises can be adopted and implemented makes them particularly promising for widespread promotion. As holistic approaches to both physical and mental health, these interventions have significant potential for improving the well-being of older adults.

Limitations

Despite the positive impact of Chinese traditional ethnic sports on sleep quality among older adults, the evidence remains of moderate quality due to several limitations. It is important to note that while all the studies included in this review were RCTs, certain limitations should be acknowledged. Firstly, the number of studies included was relatively small, and the participants may not be fully representative of the broader elderly population. This review focused specifically on the effects of traditional Chinese ethnic sports on sleep quality in older adults with insomnia as well as those in good health, limiting the generalizability of the findings to the wider elderly population. Larger studies could provide more robust evidence of our findings. Secondly, many of the included studies did not adequately control for other factors that might influence sleep quality. For instance, some studies relied on subjective self-report measures instead of more objective tools like polysomnography. Additionally, the use of hypnotic medications among participants varied, with some using them while others did not. This disparity might have impacted the reliability and consistency of the results. Additionally, both older adults with and without insomnia in the analysis may affect the intervention’s apparent effect and reduce specificity. Future studies should focus on more homogeneous samples to assess the intervention’s effects more precisely for each subgroup. Furthermore, while PSQI is a widely used metric, the exclusive reliance on self-reported data limits the findings. Future studies could incorporate objective measures, such as polysomnography, to strengthen the results. Lastly, some of the studies lacked detailed information regarding the randomization process and blinding procedures. This lack of this critical information introduces potential bias and limits the overall confidence in the conclusions of the review. Therefore, future well-designed RCTs are essential to draw more definitive conclusions about the impact of traditional Chinese ethnic sports on sleep quality in older adults.

Conclusion

Existing evidence suggests that traditional Chinese ethnic sports may have significant potential to improve sleep quality in older adults. However, the current understanding of these effects remains limited, and the available data is not yet comprehensive or entirely reliable. Future research should delve deeper into the relationship between different forms of ethnic sports and sleep quality through more rigorous RCTs. In addition, personalized and targeted intervention programs, tailored to individual differences and needs, should be developed. By integrating the promotion of traditional Chinese ethnic sports with strategies to improve sleep, the overall quality of life for older adults can be enhanced. These activities can be promoted through community centers, senior living facilities, or local health programs as part of a holistic approach to enhancing well-being. Health professionals can collaborate with local organizations to design culturally relevant and accessible exercise routines that cater to older adults’ unique physical abilities and health conditions.

Supplemental Information

Supplemental Information 1 Raw data

Supplemental Information 2 PRISMA 2020 checklist

Supplemental Information 3 Search strategies

I would like to express my sincere thanks to Professor Shiyi Cao from the School of Tongji Medical of Huazhong University of Science and Technology for his guidance and support as well as professional advice and encouragement during the whole research process.

Additional Information and Declarations

Competing Interests

Author Contributions

Data Availability

The authors declare there are no competing interests.

Jiahui Liu conceived and designed the experiments, performed the experiments, analyzed the data, prepared figures and/or tables, and approved the final draft.

Xiong-Wen Ke conceived and designed the experiments, performed the experiments, analyzed the data, authored or reviewed drafts of the article, and approved the final draft.

Yi Lan performed the experiments, analyzed the data, authored or reviewed drafts of the article, and approved the final draft.

Diana Yuan conceived and designed the experiments, prepared figures and/or tables, authored or reviewed drafts of the article, and approved the final draft.

Weihao Zhang analyzed the data, prepared figures and/or tables, and approved the final draft.

Jian Sun conceived and designed the experiments, analyzed the data, authored or reviewed drafts of the article, and approved the final draft.

The following information was supplied regarding data availability:

This is a systematic review/meta-analysis.

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
