# Peer review of "Effects of Chinese traditional ethnic sports on sleep quality among the elderly: a systematic review and meta-analysis"

_PeerJ, doi:10.7717/peerj.19019_

## Round 0.1 · original submission · Major Revisions

Dear Dr. Sun,

Thank you for your submission to PeerJ. Please address all the comments of the reviewers, in detail

·

Basic reporting

Congratulations on an interesting topic and a good article.

Experimental design

Well-chosen methods of analysis and review.

Validity of the findings

Figure 2.
The bottom part of this figure, can the authors' names be written vertically so that you don't have to turn your head?

Additional comments

Very good read of this article and analysis part of the metanalysis.

·

Basic reporting

This is a meta-analysis of Chinese traditional ethnic sports' effects on sleep quality among older adults. introduction part clearly states the problem and the role of the traditional Chinese exercising in addressing the problems .However, it lacks the description of both qigong and tai chi exercise and their techniques. In results part some of the figures are referenced but didn't analyzed in the article,

Experimental design

it is comprehensively well written including many database. Ensure figures and tables are adequately explained and integrated into the discussion.

Validity of the findings

The findings are strong in many aspects but have a bit weakness as heterogeneity that might be improved by meta-regression, and the sample size that were included from 10 articles that might be less for generalization, The inclusion of both older adults with and without insomnia in the analysis may affect the intervention's apparent effect and reduce specificity.

Additional comments

Check the language for grammatical and easier readings.

·

Basic reporting

1. Conduct a thorough proofread or professional language editing, several flaws were found during the text.

2. While the introduction provides a strong rationale, it could benefit from a clearer explanation of why Chinese traditional ethnic sports specifically are of interest. The transition from general sleep issues to these sports is abrupt.

3. While the figures are relevant, they could benefit from improved clarity in labeling. For example:
"Figure 5 Forest plot of eûects Chinese traditional entire sports"—correct the spelling and provide a more descriptive caption.

Experimental design

The study design is rigorous, adhering to PRISMA 2020 guidelines and employing appropriate meta-analytic techniques. Comprehensive database search strategy with detailed inclusion/exclusion criteria enhances reproducibility. Adequate subgroup analyses were performed to investigate moderating effects.

1. The study pools data from normal elderly individuals and those with insomnia without a strong rationale for why these populations are combined. This could introduce heterogeneity. Please, address the justification in the discussion or separate analyses for these groups.

2. While PSQI is a widely used metric, the exclusive reliance on self-reported data limits the findings. The absence of objective measures like polysomnography is a notable limitation. I suggest acknowledging this limitation more explicitly in the discussion and suggest its inclusion in future studies.

3. There is limited detail about the intervention settings (e.g., supervised or unsupervised sessions, training fidelity). I suggest expanding on how interventions were delivered, ensuring applicability.

4. While the Cochrane risk of bias tool is utilized, only a limited number of studies achieved allocation concealment and blinding. Please, discuss how this impacts the reliability of findings.

Validity of the findings

Findings are robust, supported by sensitivity analyses, and aligned with existing literature. The use of subgroup analysis adds depth to the interpretation of the results.

1. Significant heterogeneity (e.g., I² > 70% in some analyses) is observed but inadequately explored beyond sensitivity analyses. Please, provide more detailed explanations for heterogeneity sources (e.g., cultural differences, intervention intensity).

2. The study highlights that traditional Chinese ethnic sports do not improve sleep onset or daytime dysfunction, conflicting with some prior research. I suggest exploring possible reasons (e.g., population differences, intervention duration) and discuss their implications.

3. While the use of GRADEpro is appropriate, the moderate quality of evidence for most findings limits their generalizability. Please, discuss strategies to improve the quality of evidence in future research.

Additional comments

The study contributes valuable insights into non-pharmacological interventions for sleep improvement in older adults. It highlights the global relevance of traditional Chinese sports, potentially increasing cultural awareness and encouraging their adoption.

1. While the topic is relevant, the paper could better articulate its unique contributions beyond summarizing existing evidence. For instance, how does it advance understanding of sleep interventions for the elderly?

2. The conclusion could expand on practical recommendations for integrating these exercises into public health initiatives or elderly care programs.

·

Basic reporting

Line 34: Theres appear to be a typo in abstract where "s variables.leep" should be "sleep variables"
The English writting need improvement in severall sections, particulary lines 33-34, 86-89
Several referencies have inconsistant formatting (e.g., lines 61-63, 84-86)
Figure 5 is unreadible and not understandible
Some methodologic details are missing from Methods section and it is not clear if Chinese pubblications were published in journals indexed in Scopus or WOS

Experimental design

Line 102: The rationale for the 55+ age cutoff need better justification and adeguate references because I not understand the reason
The controll group interventions vary significatively between studies but this heterogeneity is not adequately addressed, also not highlighted in the limitations
The dose-responce relationship beetwen intervention duration/frequency and outcomes should be analized more systematicaly
Risk of bias assesment methodology needs more detail because I think this work is more similar to a narrative review than systematic review, there are too much areas of weakness

Validity of the findings

High heterogenity (I2 >75%) in severall analyses need more discussion
The effects of diferent types of traditional sports are not clearly differentiated and remains a large area of doubtful interpretability, I don't think the content is sufficiently accurated
The GRADE assesment suggests low quality evidence for some key outcomes, I think it would be more corect to highlight this in a more evident way

Additional comments

The Introduction would benefit from better justifcation for focusing on elderly populations and better clarify the reference to 55 years of age
More detail on search strategy and terms used and better specify the quality of Chinese journals
More comprhensive description of included studies caracteristics with better visualization of meta-analysis results. Also it is necesary more attention to thorougly address limitations
Provide more clear clinical implications and reccomendations

---

## Round 0.2 · accepted · Accept

Dear authors, we are glad to inform that the reviewers think your paper is acceptable.

·

Basic reporting

I am happy with the current version of the manuscript.

The authors did a good job of reviewing and answering all commentaries made by the reviewers.

Experimental design

I am happy with the current version of the manuscript.

The authors did a good job of reviewing and answering all commentaries made by the reviewers.

Validity of the findings

I am happy with the current version of the manuscript.

The authors did a good job of reviewing and answering all commentaries made by the reviewers.

·

Basic reporting

No comment

Experimental design

No comment

Validity of the findings

No comment

Additional comments

I have carefully reviewed the work with the suggestions I had previously indicated and the authors' modifications. I believe the authors have adequately addressed my requests and the work has improved. Although not particularly substantial or innovative, I believe the paper can be accepted in its current form.